# Injury of Corticospinal Tract in a Patient with Subarachnoid Hemorrhage as Determined by Diffusion Tensor Tractography: A Case Report

**DOI:** 10.3390/brainsci10030177

**Published:** 2020-03-19

**Authors:** Chan-Hyuk Park, Hyeong Ryu, Chang-Hwan Kim, Kyung-Lim Joa, Myeong-Ok Kim, Han-Young Jung

**Affiliations:** Department of Physical & Rehabilitation Medicine, Inha University School of medicine, Inha University Hospital, Incheon 22332, Korea; chanhyuk@gmail.com (C.-H.P.); allen121212@naver.com (H.R.); jacob.kim@inha.ac.kr (C.-H.K.); drjoakl@gmail.com (K.-L.J.); rmkmo@inha.ac.kr (M.-O.K.)

**Keywords:** subarachnoid hemorrhage, ventriculomegaly, diffusion tensor imaging, corticospinal tract

## Abstract

We report diffusion tensor tractography (DTT) of the corticospinal tract (CST) in a patient with paresis of all four limbs following subarachnoid hemorrhage (SAH) with intraventricular hemorrhage (IVH) after the rupture of an anterior communicating artery (ACoA) aneurysm rupture. The 73-year-old female was admitted to our emergency room in a semi-comatose mental state. After coil embolization—an acute SAH treatment—she was transferred to our rehabilitation department with motor weakness development, two weeks after SAH. Upon admission, she was alert but she complained of motor weakness (upper limbs: MRC 3/5, and lower limbs: MRC 1/5). Four weeks after onset, DTT showed that the bilateral CSTs failed to reach the cerebral cortex. The left CST demonstrated a wide spread of fibers within the corona radiata as well as significantly lower tract volume (TV) and higher fractional anisotropy (FA) as well as mean diffusivity (MD) compared to the controls. On the other hand, the right CST shifted to the posterior region at the corona radiata, and MD values of the right CST were significantly higher when compared to the controls. Changes in both CSTs were attributed to vasogenic edema and compression caused by untreated hydrocephalus. We demonstrate in this case, two different pathophysiological entitles, contributing to this patient’s motor weakness after SAH.

## 1. Introduction

Anterior communicating artery (ACoA) aneurysms are the most common intracranial aneurysms and account for approximately 30% to 37% of subarachnoid hemorrhage (SAH) [1]. After aneurysm rupture, the resulting subarachnoid hemorrhage can result in complications such as cognition impairment [2]. However, motor weakness is also one of the neurological complications of SAH [3].

Many studies have used diffusion tensor tractography (DTT) to visualize neural tracts in the human brain, and as a result, DTT provides a useful means of evaluating neural tract injuries in human brain [4,5,6]. However, no DTT study has been performed examining the mechanisms responsible for motor weakness in the limbs of patients during the subacute phase following SAH with intraventricular hemorrhage (IVH). We hypothesize that, based on our DTT finding, that the injury of the corticospinal tract (CST) is due to SAH with intraventricular hemorrhage (IVH).

## 2. Case Presentation

A 73-year old female patient without a relevant prior medical history was admitted to our emergency room in a semi-comatose mental status (Glasgow Coma Scale: 3). Computed tomography (CT) revealed a modified Fisher grade 4 SAH due to the rupture of a ACoA aneurysm, with consecutive hydrocephalus due to IVH (Figure 1A) [7]. After coil embolization, the patient was neurosurgically treated by hematoma evacuation and placement of an external ventricular drainage (EVD) (Figure 1B). Nimodipine was administered according to standing guidelines [8]. Two weeks after the initial ictus and upon improvement of her mental status, the patient was later transferred to our rehabilitation department. Upon admission, her mental status was altered, however, with reduced motor function of the upper and lower extremities (medical research council (MRC) grading of 3 in upper extremities and 1 in lower extremities). Bladder and bowel functions were preserved. We performed brain magnetic resonance imaging (MRI) at approximately week 4 after onset because of recovery of the CST within two weeks due to the resolution of peri-lesional edema or inflammation after stroke [9]. The result showed bilateral ventriculomegaly (Evan’s index: 0.35) with ventricular capping, encephalomalacia in both frontal lobes, but no demarcated cerebral infarctions (Figure 2A).

## 3. Diffusion Tensor Imaging

DTT images were obtained using a 3.0 T GE Signa Architect MRI System (General Electric, Milwaukee, WI, USA). The MRI preset conditions were as follows: field of view = 240 × 240 mm, acquisition matrix of 128 × 128, b = 1000 mm^2^·s^−1^, TR (repetition time) = 15,000 ms, TE (echo time) = 80.4 ms, slice thickness = 2 mm, 30 directions, and 72 contiguous slices. DTT images were analyzed using DTI studio software (www.mristudio.org, Johns Hopkins Medical Institute, Baltimore, USA). The CST was reconstructed using two regions of interest (ROIs). The seed ROIs were placed on the lower anterior pons and the target ROI was placed on the corona radiata. Termination criteria for fiber tracking were a fractional anisotropy (FA) of <0.2 and a turning angle of >60°. Mean diffusivities (MDs), tract volumes (TVs), and FAs of the CST tracks were measured. Results were compared with seven age-matched healthy control subjects (two males with a mean age of 70.14 years, age range 67–78 years) in this study. In prior analysis, we defined pathological changes as a deviation from reference values with at least two standard deviations [10]. At four weeks after onset, DTT revealed curved changes of both CSTs around both lateral ventricles due to bilateral ventriculomegaly compared with control subjects (Figure 2B). Furthermore, the right CST was shifted to the posterior region at the corona radiata without reaching the cerebral cortex, which was called the discontinuation (Figure 2B,C). The left CST appeared more spread out compared to the controls when looking at subcortical white matter and showed only a few fibers reaching the cerebral cortex (Figure 2B,C,E). FA and TV values of the left CST and MD values of both CSTs exhibited differences of two (SD) compared to our reference values (Table 1).

## 4. Discussion

The corticospinal tract (CST) constitutes the main white matter motor pathway [11]. In this patient with generalized motor weakness after SAH, the CST was investigated using DTT MRI imaging. Bilateral CST reconstruction using DTT revealed side-dependent differences between our subject in comparison to a matched cohort of control patients. More specifically, changes in CST structures were observed around the corona radiata and lateral ventricle; the right CST had moved posteriorly and the left CST was spread out at the level of the corona radiata, compared to the healthy controls. Apart from displacement, the CSTs revealed signs of discontinuation at the cerebral cortex. Furthermore, in the left CST, the FA and MD values were elevated and TV values reduced, and in the right CST, the MD values were alerted due to a greater or lower difference than two standard deviations compared with the control subjects.

FA values represent degrees of directionality at a microscopic level [10,12,13], and thus provide a means of assessing microstructural integrity of axons, myelin, and microtubules [4,14]. On the other hand, MD values provide quantitative measures of water diffusion, and are indicative of pathological changes taking place in white matter [15]. Increases in the values of MD represent vasogenic edema or axonal damage whereas a decrease in MD values indicates neural injury [16]. TV values represent numbers of voxels within neural tracts [16]. Therefore, FA and TV reductions in combination with MD increases indicate the presence of neural injury [6,16].

In a previous study, it was reported that patients with normal pressure hydrocephalus (NPH) had initial high FA values, explained by mechanical compression and decreased FA and increased MD values as a result of secondary degenerative changes [3,17,18]. In this case, our patient exhibited bilateral displacement and discontinuation of CSTs before reaching the cerebral cortex. In addition, our MR imaging revealed high FA and MD values in combination with low TV value in the left CST. However, the FA value of the right CST was not increased significantly. Based on these findings, we hypothesized that the mechanism of motor weakness in our case was different between sides. At week 4 after onset, posterior displacement of the right CST, as the result of hydrocephalus caused ventriculomegaly, resulted in a discontinuation of the tract to the cerebral cortex. Significantly high MD values were found with the number of voxels contained within neural tracts and FA values remaining unchanged. We believed that the changes to the right CST are mainly caused by vasogenic edema without neural injury, leading to the discontinuation of the tract resulting loss of left motor function. In contrast to our results during the subacute phage, a previous study reported the formation of vasogenic edema following SAH in the acute phase [19]. Further MRI assessment of the timely development of vasogenic edema after SAH is necessary. However, in the left CST, spreading within the corona radiata was observed due to ventriculomegaly caused mechanical compression. This, based on the observed increased FA values, decreased TV, and increased MD indicative of degenerative change or neural tract injury. These observations led to the paralysis of right limbs. However, we were able to exclude age-associated microstructural changes because while age-associated microstructural changes using DTT showed increased FA and MD values, our patient reveal no reduced FA value [20]. Additionally, as a previous study, the analysis of CST for upper and lower limbs using DTT is difficult due to the discontinuation of CSTs at the cerebral cortex [21]. However, we believe that a few fibers to the cerebral cortex could play a role in upper motor function and tracts associated with lower motor function were injured due to a compression of lateral ventricle and their discontinuation. However, the defined causes of mechanical compression on the left side there and vasogenic edema in the right side were not defined as effects at four weeks after SAH, so further study is required.

Therefore, at four weeks after onset (subacute phase), injury of the left tracts resulted from degenerative change of the tracts (increased MD value and decreased TV value) and mechanical compression (increased FA value). Damage to the right CST caused vasogenic edema by ventriculomegaly. We believe that because of the larger left ventricle, mechanical compression had greater influence on the left CST in terms of increasing FA. In other words, we indicate that mechanical compression and degenerative changes by ventriculomegaly can coexist and induce tract injury. In addition, we believe that the right CST injury resulted from vasogenic edema based on differences in morphology of the tract caused by the difference in ventricle size.

The most feared complication following SAH is delayed cerebral ischemia. To prevent delayed cerebral ischemia, our patient administrated aspirin and cilostazol [8]. Another complication is hydrocephalus [7]. The patient received an EVD during the acute phase and was constantly being monitored for persisting hydrocephalus. Previous studies have also presented mechanisms of motor weakness after SAH or NPH [3,17]. It has been suggested that one of the mechanisms of paraplegia or paraparesis after SAH is hydrocephalus [3]. However, to the best of our knowledge, this case report is the first to observe a side-dependent different mechanism resulting in motor weakness as a result of asymmetrical hydrocephalic ventriculomegaly (mechanical compression at the left hemisphere and vasogenic edema at the right hemisphere). However, this study has some limitations, as follows: (1) The external validity of these observations remains limited because they are based on a single case, larger scale long-term studies are necessary to confirm our hypothesis; (2) As our study was in the subacute phase, long-term follow up is necessary; (3) DTT interpretation is operator-dependent, leading to potential performance bias [11]; and (4) Finally, further clarification via electrophysiological examination of long fiber tracts would have been helpful, but was hindered due to the existing craniotomy defect [22].

## 5. Conclusions

Although DTT is subjected to operator-dependency [11], we can conclude a coexistence of side-dependent compressive and degenerative damage to the CST caused by either ventriculomegaly or direct compression, leading to bilateral motor weakness. Here, DTT was proven to be a useful tool in assessing the mechanism behind post-SAH persisting motor weakness.

## Figures and Tables

**Figure 1 brainsci-10-00177-f001:**
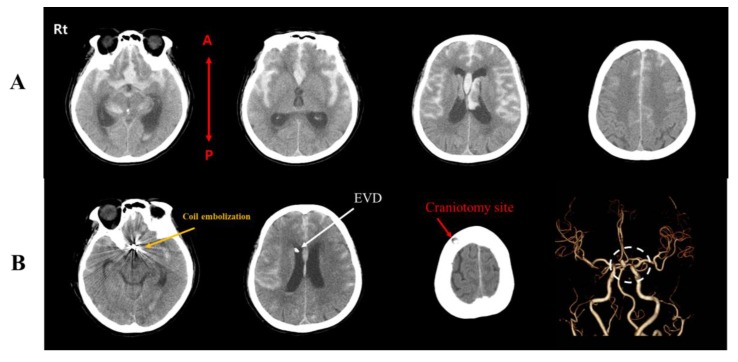
(**A**) Initial computed tomography (CT) images showing subarachnoid hemorrhage (SAH) with intraventricular hemorrhage (IVH) and consecutive ventriculomegaly, resulting from a ruptured anterior communicating artery aneurysm (ACoA). (**B**) CT image after coil embolization and external ventricular drainage (EVD) placement. Coil embolization (yellow arrow), EVD (white arrow), craniotomy site at the right frontal lobe (red arrow), and ACoA (white dashed circle). Note: EVD; external ventricular drainage, R, right; L, left; A, anterior; P, posterior.

**Figure 2 brainsci-10-00177-f002:**
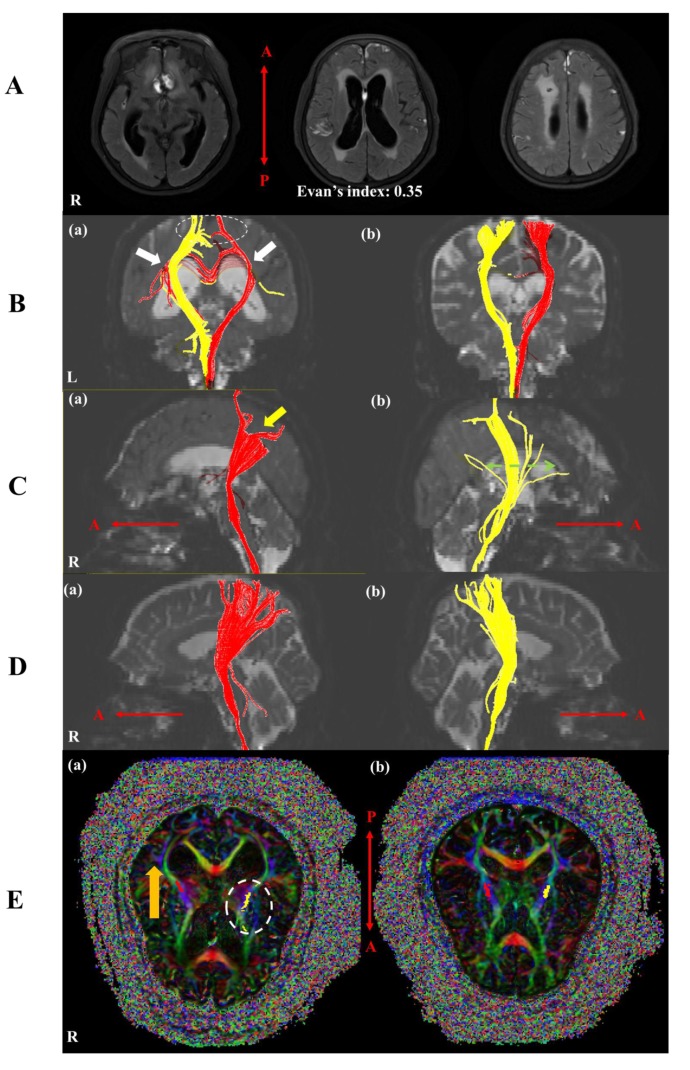
(**A**) Brain magnetic resonance images obtained at week 4 after onset showing clearance of subarachnoid blood with persisting bilateral ventriculomegaly (Evan’s index: 0.35) with interstitial edema and encephalomalacia in both frontal lobes (severe enlargement of the left ventricle). (**B**) (a) Changes in both CSTs were observed around both lateral ventricles due to bilateral ventriculomegaly versus the control subjects (70 years old female, white arrow) and a discontinuity of both tracts to the bilateral cortex was apparent (white dash circle) (b) control. (**C**) (a) The right CST did not extend to the right cortex (yellow arrow) and (b) the left CST spread out (green dash arrow). (**D**) CSTs of a control subject. (**E**) (a) The right CST (red) was posteriorly shifted at the corona radiata (orange arrow) and the left CST (yellow) was spread out compared with control subjects (white dashed circle). (b) Left and right CSTs of a control subject. Note: R, right; L, left; A, anterior; P, posterior.

**Table 1 brainsci-10-00177-t001:** Diffusion tensor tractography (DTT) parameter values of the corticospinal tracts of the patient and control subjects.

		FA	TV	MD (×10^−3^ mm^2^/s)
Patient	Right	0.632	3106.000	0.879 **
	Left	0.666 **	2148.000 **	0.763 **
Controls (*n* = 7)	Subject 1 (F/78)	0.617	3211.500	0.714
	Subject 2 (F/70)	0.598	2703.500	0.712
	Subject 3 (M/67)	0.623	2704.500	0.691
	Subject 4 (F/68)	0.612	2998.500	0.717
	Subject 5 (M/70)	0.633	3150.000	0.694
	Subject 6 (F/68)	0.624	3203.500	0.724
	Subject 7 (F/70)	0.607	3047.000	0.715
	Mean (SD)	0.616 (0.017)	3002.643 (218.341)	0.709 (0.012)

SD, standard deviation; F, female; M, male; CST, corticospinal tract; FA, fractional anisotropy; TV, tract volume; MD, mean diffusivity. ** Parameters were two SDs above or below mean normal control subject values.

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
