# Peer review of "Injury of Corticospinal Tract in a Patient with Subarachnoid Hemorrhage as Determined by Diffusion Tensor Tractography: A Case Report"

_brainsci, 2020, doi:10.3390/brainsci10030177_

Round 1

Reviewer 1 Report

The authors present the case of a 73-year old SAH patients after rupture of an ACoA aneurysm. Based on the clinical description the hemorrhage can be described as a  Hunt and Hess grade 4 bleed, mFisher 4 on CT-imaging. 
During rehabilitation, approximately 4 weeks after the initial ictus, an MRI scan was done using tractography of the corticospinal tract. The patient presented at the time with a generalized weakness of the upper extremities (MRC 3/5) and paraplegia (MRC 1/). The MRI image showed in our opinion a marked and untreated post-hemorrhagic hydrocephalus with ventricular capping and generalized diapedesis of CSF. We believe it to be a far cry to interpret the interhemispheric differences in fractional anisotropy and mean diffusivity along the corticospinal tract as having different etiologies. What would be the cause of the left-sided "mechanical compression" and the "vasogenic" right-sided edema. The underlying pathology is a non-obstructive hydrocephalus and in our opinion, DTI results can only be valuably be interpreted after CSF shunting. 

Author Response

1) The MRI image showed in our opinion a marked and untreated post-hemorrhagic hydrocephalus with ventricular capping and generalized diapedesis of CSF..

Ans> Thank you for your great comment.

The patient underwent coil embolization and craniotomy for external ventricular drainage (EVD). Using EVD, her mental status was checked. When her mental status was alert, EVD was removed. So we revised that Figure 1 added EVD status CT image, craniotomy site image, and coil embolization image and the sentence as follows;

Line 42-47: The patient underwent coil embolization and craniotomy for external ventricular drainage (EVD) hematoma evacuation on Kocher's point the right frontal lobe at the department of neurosurgery (Figure 1B). After surgery, EVD was controlled, and nimodipine was administrated. When her mental was altered, and she was later transferred with EVD and nimodpine removal to our rehabilitation department at 2 weeks after onset.

Line 158-161: (2) Because our study was in the subacute phase, long-term follow up is necessary. Additionally, the long term image follows up for the determination of motor recovery and the change of ventricle size for hydrocephalus is being required.

Figure 1. (A) Initial computed tomography (CT) images showing SAH with ventriculomegaly caused by intraventricular hemorrhage (IVH) resulting from a ruptured anterior communicating artery (ACoA). (B) CT image after coil embolization and EVD. Coil embolization (yellow arrow), EVD (white arrow), craniotomy site at the right frontal lobe (red arrow), and ACoA (white dashed circle). Note: EVD; external ventricular drainage, R, right; L, left; A, anterior; P, posterior.

2) We believe it to be a far cry to interpret the interhemispheric differences in fractional anisotropy and mean diffusivity along the corticospinal tract as having different etiologies. What would be the cause of the left-sided "mechanical compression" and the "vasogenic" right-sided edema. The underlying pathology is a non-obstructive hydrocephalus and in our opinion, DTI results can only be valuably be interpreted after CSF shunting.

Ans> your comment is right. Thank you for your great comment. As your mention, I thought that the importance is the cause of the left-sided "mechanical compression" and the "vasogenic" right-sided edema. However, this report is case report, and the definition about mechanism and cause is so difficult. Thus, we revised discussion as follows;

Line 161: As the above mention, because causes of mechanical compression in the left side and vasogenic edema were not defined, the further study using large scale is required. (4) DTT depends on handling of the operator. [8].

Line 138: However, the defined causes of mechanical compression in the left side there and vasogenic edema in the right side were not defined effect at 4 weeks after SAH,. the further study was required.

Also

To address Statistical

Line 67-68 Add “we used statistical analysis suggested in the previous study that it was abnormal that DTT parameter values (FA, MD, and TV values) indicated more or lower difference than two SDs (standard deviations) compared with control subjects [11].”

Reviewer 2 Report

The authors present the case of a 73 year old female SAH (subarachnoid hemorrhage) patient. She underwent coiling of an Acom aneurysm, furthermore a craniotomy was performed. After 2 weeks she was transferred to the rehabilitation department. She exhibited a tetraparesis. MRI imaging showed ventriculomegaly with interstitial edema and encephalomalacia in the frontal lobes and lesions in the CSTs as most likely reason for the tetraparesis. The authors conclude that motor weakness was caused by CST lesions caused by compression and degenerative changes caused by ventriculomegaly following SAH.

I have some aspects that need to be addressed:

  • What was the indication for craniotomy after aneurysm coiling? Where was the craniotomy performed? Could the lesions be partly related to the surgery?
  • How was hydrocephalus treated? Did the patient have a ventricular or lumbar drain for treatment of hydrocephalus?
  • Did the patient undergo electrophysiological diagnostics?
  • In fact, a relatively large number of SAH patients develops brain lesions related to delayed cerebral ischemia, which do not necessarily relate to a vessel territory (e.g.: Nat Rev Neurol. 2014 Jan;10(1):44-58). How can the authors exclude that the lesions are DCI-related? This aspect should be included into the discussion.
  • Indications for hydrocephalus treatment in SAH patients are under debate. Does the case presented here indicate that hydrocephalus after SAH should be treated more aggressively?

Author Response

1) What was the indication for craniotomy after aneurysm coiling? Where was the craniotomy performed? Could the lesions be partly related to the surgery?

Ans>

Line 43-47: Craniotomy was performed for EVD (external ventricular drainage on frontal lobe.

revised sentenceà The patient underwent coil embolization and craniotomy for external ventricular drainage (EVD) on Kocher's point the right frontal lobe at the department of neurosurgery (Figure 1B). After surgery, EVD was controlled, and nimodipine was administrated. When her mental was altered, she was transferred with EVD and nimodpine removal to our rehabilitation department at 2 weeks after onset.

Figure 1: we add figures including EVD state, coli embolization, craniotomy site in Figure

revised sentenceà Figure 1. (A) Initial Ccomputed tomography (CT) images showing SAH with ventriculomegaly caused by intraventricular hemorrhage (IVH) resulting from a ruptured anterior communicating artery (ACoA). (B) CT image after coil embolization and EVD. Coil embolization (yellow arrow), EVD (white arrow), craniotomy site at the right frontal lobe (red arrow), and ACoA (white dashed circle). Note: EVD; external ventricular drainage, R, right; L, left; A, anterior; P, posterior.

2) How was hydrocephalus treated? Did the patient have a ventricular or lumbar drain for treatment of hydrocephalus?

Ans> Referred to your comment, we revised sentence about treatments that the patient underwent

Line 43-47: the patient performed EVD for CSF drainage, and when the patient mental status was alert, EVD was removed.

Line 160: Because our study was in the subacute phase, long-term follow up is necessary. Additionally, the long term image follows up for the determination of motor recovery and the change of ventricle size for hydrocephalus is being required

3) Did the patient undergo electrophysiological diagnostics?

Ans> your comment was right. Thus, we change the following

Line 163-164: we add the sentence and reference “ [21]”as follows:

(5) The patient did not undergo electrophysiological diagnostics for motor pathways due to skull defect by craniotomy [21].

4) In fact, a relatively large number of SAH patients develops brain lesions related to delayed cerebral ischemia, which do not necessarily relate to a vessel territory (e.g.: Nat Rev Neurol. 2014 Jan;10(1):44-58). How can the authors exclude that the lesions are DCI-related? This aspect should be included into the discussion.

Ans>

Thanks your respectable comment. We change the following

Line 45-46: We add the sentence “After surgery, EVD was controlled, and nimodipine was administrated’

Line 50-51: we add the sentence “At the transferred time, she took memantine for cognitive impairment, calcium channel blocker for blood pressure, and antiplatelet medications (aspirin and cilostazol).”

Line 54-55: add “and did not observe cerebral infarction ”

Line 149-152: we demonstrated complications following SAH, as well as drugs for DCI and EVD for hydrocephalus were added

“ The representative complication following SAH is delayed cerebral ischemia. Thus, to prevent delayed cerebral ischemia, our patient administrated aspirin and cilostazol [22]. Another complication is hydrocephalus [23]. The patient performed EVD at acute phase and constantly being monitored for hydrocephalus”

5) “Indications for hydrocephalus treatment in SAH patients are under debate. Does the case presented here indicate that hydrocephalus after SAH should be treated more aggressively?

Ans> Your comment was right. However, phase of our case report was subacute phase, and patient underwent EVD for hydrocephalus. After that until now, her neurologic change and brain image are being followed up.

Line 158-159: (2) Because our study was in the subacute phase, long-term follow up is necessary. Additionally, the long term image follows up for the determination of motor recovery and the change of ventricle size for hydrocephalus is being required

Also

To address Statistical

Line 67-68 Add “we used statistical analysis suggested in the previous study that it was abnormal that DTT parameter values (FA, MD, and TV values) indicated more or lower difference than two SDs (standard deviations) compared with control subjects [11].”

Round 2

Reviewer 1 Report

Please consider following changes to the text:

Abstract: We report diffusion tensor tractography (DTT) of the corticospinal tract (CST) in a patient with paresis of all four limbs following subarachnoid hemorrhage (SAH) with intraventricular hemorrhage (IVH) after the rupture of an anterior communicating artery (ACoA) aneurysm. The 73-year-old 13 female was admitted to our emergency room in a semi-comatose state. After coil embolization an acute SAH treatment, she was transferred to our rehabilitation department with motor weakness development, two weeks after SAH. Upon admission, she was alert but  complained of motor weakness (upper limbs: MRC 3/5, and lower limbs: MRC 1/5). Four weeks after onset, DTT showed the bilateral CSTs failed to reach the cerebral cortex. The left CST demonstrated a wide spread of fibers within the corona radiata as well as significantly lower tract volume (TV), higher fractional anisotropy (FA) and mean diffusivity (MD) compared to controls. On the other hand, the right CST shifted to the posterior region at the corona radiata, and MD values of the right CST were significantly higher compared to controls. Changes in both CSTs were attributed to vasogenic edema and compression caused by untreated hydrocephalus. We demonstrate in this case, two different pathophysiological entities, contributing to this patients motor weakness after SAH.

Keywords: subarachnoid hemorrhage; ventriculomegaly; diffusion tensor imaging; corticospinal tract

  1. Introduction 
    Anterior communicating artery (ACoA) aneurysms are the most common intracranial aneurysms and account for approximately 30 to 37% of SAH cases. After aneurysm rupture, the resulting subarachnoid hemorrhage can result in long-term complications such as cognition impairment [2]. 31 However, motor weakness is also one of the neurological complications of SAH [3].
    Many studies have used diffusion tensor tractography (DTT) to visualize neural tracts in the human brain, and as a result, DTT provides a useful means of evaluating neural tract injuries [4–6]. However, no DTT study has been performed examining the mechanisms responsible for motor weakness in the limbs of patients during the subacute phase following SAH with IVH. We hypothesize that, based on our DTT finding, that the injury of the corticospinal tract (CST) is due to SAH caused intraventricular hemorrhage (IVH). 

  2. 2. Case presentation
    A 73-year old female patient without a relevant prior medical history was admitted to our emergency room in a semi-comatose mental status (Please report initial GCS). Computed tomography (CT) revealed a modified Fisher grade 4 SAH due to the rupture of a ACoA aneurysm, with consecutive hydrocephalus due to IVH.  (Figure 1A) [7]. After coil embolization, the patient was neurosurgically treated by hematoma evacuation and placement of an external ventricular drainage (EVD) (Figure 1B). Nimodipine was administered according to standing guidelines. (Reference). Two weeks after the initial ictus and upon improval of her mental the patient was later transferred to our rehabilitation department. Upon admission, her mental status was alert, however with reduced motor function of the upper and lower extremities (medical research council (MRC) grading of 3 in upper extremities and 1 in lower extremities). Bladder and bowel functions were preserved. At the transferred time, she took memantine for cognitive impairment, 50 calcium channel blocker for blood pressure, and antiplatelet medications (aspirin and cilostazol). (not relevant) (What triggered this imaging after 4 weeks, when the paraparesis was already present after 2 weeks?) Brain magnetic resonance imaging (MRI) at week 4 after onset showed bilateral ventriculomegaly (Evan’s index: 0.35) with ventricular capping, encephalomalacia in both frontal lobes but no demarcated cerebral infarctions.  (along the external ventricular drain and SAH resolution, and did not observe cerebral 54 infarction ) (not relevant) (Figure 2A). 
    3. Diffusion Tensor Imaging 
    DTT images was performed using a 3.0 T GE Signa Architect MRI System (General Electric, 57 Milwaukee, WI, United StatesUSA). (MRI presets: field of view = 240 mm × 240 58 mm2, acquisition matrix of 128 x 128 matrix, b = 1000 mm2s-1, TR (repetition time) = 15,000 ms, TE 59 (echo time) = 80.4 ms, slice thickness = 2 mm, 30 directions, and 72 contiguous slices). DTT images were analyzed using DTI studio software (www.mristudio.org, Johns Hopkins Medical Institute, 61 Baltimore, MD). The CST was reconstructed using two regions of interest (ROIs). The seed ROIs were placed on the lower anterior pons and the target ROI was placed on the corona radiata. Termination criteria for fiber tracking were a fractional anisotropy (FA) of < 0.2 and a turning angle of > 60°. Mean diffusivities (MDs), tract volumes (TVs) and FAs of CST tracks were measured. Results were compared with 7 age-matched healthy control subjects (2 males with mean 66 age of 70.14 years, age range 67-78 years) in this study (Please provide a small table depicting relevant criteria for groups matching with the corresponding values for the control group). In prior analysis we defined pathological changes as a deviation from reference values with at least 2 standard deviations. [11]. At week 4 after onset, DTT revealed curved changes of both CSTs around both lateral ventricles due to bilateral ventriculomegaly as compared with control subjects (Figure 2B). Furthermore, the right CST was shifted to the posterior region at the corona radiata and without reaching the cerebral cortex. (Figure 2B, C). The left CST appeared more spread out compared to controls when looking at subcortical white matter and showed only few fibers reaching the cerebral cortex (Figures 2B, C, and E). FA and TV values of the left CST and MD values of both CSTs exhibited differences of 2 standard deviations 75 (SD) compared to our reference values (Table 1). 
    Figure 1. (A) Initial Computed tomography (CT) images showing SAH with intraventricular hemorrhage and consecutive ventriculomegaly resulting from a ruptured anterior communicating artery aneurysm (ACoA). (B) CT image after coil embolization and EVD placement. Coil embolization (yellow arrow), EVD (white arrow), craniotomy site at the right frontal lobe (red arrow), and ACoA (red arrow)white dashed circle). Note: EVD; external ventricular drainage, R, right; L, left; A, anterior; P, posterior. 83

    Figure 2. (A) Brain magnetic resonance images obtained at week 4 after onset showing clearance of subarachnoid blood with persisting bilateral ventriculomegaly (Evan’s index: 0.35), interstitial edema, encephalomalacia in both frontal lobes (severe enlargement of the left ventricle).
  3. (B) (a) Changes in both CSTs were observed around both lateral ventricles due to bilateral ventriculomegaly versus control subjects (70
    89 years old female, white arrow) and a discontinuity of both tracts to the bilateral cortex was apperent
    (white dash circle) (b) control. (C) (a) The right CST did not extend to the right cortex (yellow arrow)
    and (b) the left CST spread out (green dash arrow). (D) CSTs of a control subject. (E) (a) The right CST
    (red) was posteriorly shifted at the corona radiata (orange arrow) and the left CST (yellow) was spread out as compared with control subjects (white dashed circle). (b) Left and right CSTs of a control
    subject. Note: R, right; L, left; A, anterior; P, posterior.
    95 Table 1. DTT parameter values of the corticospinal tracts of the patient and control subjects. Note: SD:, standard deviation,; CST, corticospinal tract; FA, fractional anisotropy; TV, tract volume; MD, mean diffusivity. **Parameters were two SDs above or below mean normal control subject values.
    Discussion
    The corticospinal tract constitutes the main white matter motor pathway  [8]. In this patient with generalized motor weakness after SAH, the CST was investigated using DTI/DTT MR imaging. Bilateral CSTs reconstruction using DTT revealed side-dependent differences between are subject in comparison to a matched cohort of control patients.  More specifically, changes in CST structures were observed around the corona radiata and lateral ventricle; the right CST had moved posteriorly and
     the left CST was spread out at the level of the corona radiata, compared to healthy controls. Apart from displacement, the CSTs revealed signs of discontinuation at the cerebral cortex. Furthermore, in the left CST, FA and
    MD values were significantly elevated and TV values significantly reduced, and in the right CST, MD values were significant alerted (in what manner?).
    FA values represent degrees of directionality at a microscopic level [9–11], and thus, provide a means of assessing microstructural integrity of axons, myelin, and microtubules [4,12]. On the other hand, MD values provide quantitative measures of water diffusion, and are are indicative for pathological changes taking place in white matter [13]. Increases in MD represents vasogenic edema or axonal damage whereas a decrease of MD values indicate neural injury [19]. TV values represent numbers of voxels within neural tracts [14]. Therefore, FA and TV reductions in combination with a MD increases, indicate the presence of neural injury [6,14].
    In a previous study, it was reported that patients with normal pressure hydrocephalus (NPH)
    had initial high FA values, explained by mechanical compression and decreased FA and increased MD values as a result of secondary degenerative changes.  [3,15,16]. In this case, our patient exhibited the bilateral displacement and discontinuation of the CSTs before reaching the cerebral cortex. In addition, our MR imaging revealed high FA and MD values in combination with low TV value in the left CST. The FA value of the right CST however, was not increased significantly. Based on these findings, we hypothesize the mechanism of motor weakness in our case to be different between sides. At week 4 after onset, posterior displacement of the right CST, as the result of hydrocephalus caused ventriculomegaly, resulted in a discontinuation of the tract to the cerebral cortex. This while significantly high MD values were found with the number of voxels contained within neural tracts
    and FA values remaining unchanged. We believe that the changes to the right CST are mainly caused by vasogenic edema without neural injury, leading to the discontinuation of the tract resulting in loss of left motor function. In contrast to our results during the subacute phase, a previous study reported the formation of vasogenic edema following SAH in the acute phase [20]. Further MRI assessment of the timely development of vasogenic edema after SAH is necessary. 
    However, in the left CST, a spreading within the corona radiata was observed due to ventriculomegaly caused mechanical compression. This, based on the observed increased FA values, decreased TV
    and increased MD indicative of degenerative change or neural tract injury. These observations lead to the paralysis of right limbs.
  4.  
  5. However, we were able to exclude age-
    FA TV MD ( ⅹ 10-3mm2/s)
  6. (please clarify this age correction further)

  7. Patient
    Right 0.632 3106.000 0.879**
    Left 0.666** 2148.000** 0.763**
    Controls mean (SD) (n=7) 0.616 (0.017) 3002.286 (451.384) 0.709 (0.019)
    Formatted: Indent: Left: 0 cm, Right: 0 cm

    associated microstructural changes because of no reduced FA value in a patient [17].
  8. This next section tries to explain the side difference in motor function, however it is unclearly written and the exact message does not come across. 
  9. Additionally, regardless of the different motor power between upper and lower limbs, the exact anatomic location analysis of CST using DTT is difficult due to the discontinuation of CSTs at the cerebral cortex [18]. However, we believe that a few fibers to the cerebral cortex could play a role in upper motor function and tracts associated with lower motor function were injured due to a compression of lateral ventricle and their discontinuation. For However, the defined causes of mechanical compression on the left side there and vasogenic edema in the right side were not defined effect at 4 weeks after SAH,. the further study was required. Therefore, at 4 four weeks after onset (subacute phase), injury of the left tracts resulted from 140 degenerative change of tracts (increased MD value and decreased TV value) and still mechanical compression (increased FA value). Damage to the right CST caused vasogenic edema by ventriculomegaly. We believe that because of the larger left ventricle, mechanical compression had greater influence on the left CST in terms of increasing FA. In other words, we indicate that mechanical compression and degenerative changes by ventriculomegaly can coexist and induce tract injury. In addition, we believe that the right CST injury results from vasogenic edema based on differences in morphology of the tract caused by difference in ventricular size.
    The most feared complication following SAH is delayed cerebral ischemia. To prevent delayed cerebral ischemia, our patient was treated with aspirin and cilostazol [22]. Another complication is hydrocephalus [23]. The patient received an EVD during the acute phase and was constantly being monitored for persisting hydrocephalus. Previous studies have also presented mechanisms of motor weakness after SAH or NPH [3,15]. It has been suggested that one of mechanisms of paraplegia or paraparesis after SAH was hydrocephalus [3]. However, to the best of our knowledge, this case report is the first to observe side-dependent different mechanisms resulting in motor weakness as a result of asymmetrical hydrocephalic ventriculomegaly (mechanical compression at the  left hemisphere and vasogenic edema at the right hemisphere). However, this study has some limitations including: (1)
  10. The external validity of these observation remains limited because they are based on a single case, demanding larger scale long-term studies are to confirm our hypothesis (2) because our study was in the subacute phase, long-term follow up is necessary. Additionally, the long-term image follows up for the determination of motor recovery and the change of ventricle size for hydrocephalus is being required (what does this sentence try to say?); (3) as the above mention, because causes of mechanical compression in the left side and vasogenic edema were not defined, the further study using large scale is required; (this is repetition of the same argument) (4) DTT interpretation is operator-dependent leading to potential performance bias [8]; and (5) Finally, further clarification via eletrophysiological examination of long fiber tracts would have been helpful but was hindered due to the existing craniotomy defect [21].  

    5. Conclusion
    Although DTI is subjected to operator-dependency [8], we observed the coexistence of side-dependent compressive and degenerative damage to the CST cause by either ventriculomegaly or direct compression leading to bilateral motor weakness. Here, DTT proves to be a useful tool in assessing the mechanism behind post-SAH persisting motor weakness. 

Author Response

Abstract: We report diffusion tensor tractography (DTT) of the corticospinal tract (CST) in a patient with paresis of all four limbs following subarachnoid hemorrhage (SAH) with intraventricular hemorrhage (IVH) after the rupture of an anterior communicating artery (ACoA) aneurysm. The 73-year-old 13 female was admitted to our emergency room in a semi-comatose state. After coil embolization an acute SAH treatment, she was transferred to our rehabilitation department with motor weakness development, two weeks after SAH. Upon admission, she was alert but  complained of motor weakness (upper limbs: MRC 3/5, and lower limbs: MRC 1/5). Four weeks after onset, DTT showed the bilateral CSTs failed to reach the cerebral cortex. The left CST demonstrated a wide spread of fibers within the corona radiata as well as significantly lower tract volume (TV), higher fractional anisotropy (FA) and mean diffusivity (MD) compared to controls. On the other hand, the right CST shifted to the posterior region at the corona radiata, and MD values of the right CST were significantly higher compared to controls. Changes in both CSTs were attributed to vasogenic edema and compression caused by untreated hydrocephalus. We demonstrate in this case, two different pathophysiological entities, contributing to this patients motor weakness after SAH.

Keywords: subarachnoid hemorrhage; ventriculomegaly; diffusion tensor imaging; corticospinal tract

Ans> as your comment, we revised manuscript.

Introduction 
Anterior communicating artery (ACoA) aneurysms are the most common intracranial aneurysmsand account for approximately 30 to 37% of SAH After aneurysm rupture, the resulting subarachnoid hemorrhage can result in long-term complications such as cognition impairment [2]. 31 However, motor weakness is also one of the neurological complications of SAH [3].
Many studies have used diffusion tensor tractography (DTT) to visualize neural tracts in the human brain, and as a result, DTT provides a useful means of evaluating neural tract injuries [4–6]. However, no DTT study has been performed examining the mechanisms responsible for motor weakness in the limbs of patients during the subacute phase following SAH with IVH. We hypothesize that, based on our DTT finding, that the injury of the corticospinal tract (CST) is due to SAH caused intraventricular hemorrhage (IVH). 

Ans> as your comment, we revised manuscript.

Case presentation
A 73-year old female patient without a relevant prior medical history was admitted to our emergency room in a semi-comatose mental status (Please report initial GCS). Computed tomography (CT) revealed a modified Fisher grade 4 SAH due to the rupture of a ACoA aneurysm, with consecutive hydrocephalus due to IVH.  (Figure 1A) [7]. After coil embolization, the patient was neurosurgically treated by hematoma evacuation and placement of an external ventricular drainage (EVD) (Figure 1B). Nimodipine was administered according to standing guidelines. (Reference).Two weeks after the initial ictus and upon improval of her mental the patient was later transferred to our rehabilitation department. Upon admission, her mental status was alert, however with reduced motor function of the upper and lower extremities (medical research council (MRC) grading of 3 in upper extremities and 1 in lower extremities). Bladder and bowel functions were preserved. At the transferred time, she took memantine for cognitive impairment, 50 calcium channel blocker for blood pressure, and antiplatelet medications (aspirin and cilostazol). (not relevant) (What triggered this imaging after 4 weeks, when the paraparesis was already present after 2 weeks?) Brain magnetic resonance imaging (MRI) at week 4 after onset showed bilateral ventriculomegaly (Evan’s index: 0.35) with ventricular capping, encephalomalacia in both frontal lobes but no demarcated cerebral infarctions.  (along the external ventricular drain and SAH resolution, and did not observe cerebral 54 infarction ) (not relevant) (Figure 2A). 

Ans> as your comment, we revised manuscript

Diffusion Tensor Imaging 
DTT images was performed using a 3.0 T GE Signa Architect MRI System (General Electric, 57 Milwaukee, WI, United StatesUSA). (MRI presets: field of view = 240 mm × 240 58 mm2, acquisition matrix of 128 x 128 matrix, b = 1000 mm2s-1, TR (repetition time) = 15,000 ms, TE 59 (echo time) = 80.4 ms, slice thickness = 2 mm, 30 directions, and 72 contiguous slices). DTT images were analyzed using DTI studio software (www.mristudio.org, Johns Hopkins Medical Institute, 61 Baltimore, MD). The CST was reconstructed using two regions of interest (ROIs). The seed ROIs were placed on the lower anterior pons and the target ROI was placed on the corona radiata. Termination criteria for fiber tracking were a fractional anisotropy (FA) of < 0.2 and a turning angle of > 60°. Mean diffusivities (MDs), tract volumes (TVs) and FAs of CST tracks were measured. Results were compared with 7 age-matched healthy control subjects (2 males with mean 66 age of 70.14 years, age range 67-78 years) in this study (Please provide a small table depicting relevant criteria for groups matching with the corresponding values for the control group). In prior analysis we defined pathological changes as a deviation from reference values with at least 2 standard deviations. [11]. At week 4 after onset, DTT revealed curved changes of both CSTs around both lateral ventricles due to bilateral ventriculomegaly as compared with control subjects (Figure 2B). Furthermore, the right CST was shifted to the posterior region at the corona radiata and without reaching the cerebral cortex. (Figure 2B, C). The left CST appeared more spread out compared to controls when looking at subcortical white matter and showed only few fibers reaching the cerebral cortex (Figures 2B, C, and E). FA and TV values of the left CST and MD values of both CSTs exhibited differences of 2 standard deviations 75 (SD) compared to our reference values (Table 1)

Ans> as your comment, we revised manuscript,

Ans> Please provide a small table depicting relevant criteria for groups matching with the corresponding values for the control group

-->à we edited Table 1.

Figure 1. (A) Initial Computed tomography (CT) images showing SAH with intraventricular hemorrhage and consecutive ventriculomegaly resulting from a ruptured anterior communicating artery aneurysm (ACoA). (B) CT image after coil embolization and EVD placement. Coil embolization (yellow arrow), EVD (white arrow), craniotomy site at the right frontal lobe (red arrow), and ACoA (red arrow)white dashed circle). Note: EVD; external ventricular drainage, R, right; L, left; A, anterior; P, posterior. 83
 Ans> as your comment, we revised manuscript,

Figure 2. (A) Brain magnetic resonance images obtained at week 4 after onset showing clearance of subarachnoid bloodwith persistingbilateral ventriculomegaly (Evan’s index: 0.35), interstitial edema, encephalomalacia in both frontal lobes (severe enlargement of the left ventricle).(B) (a) Changes in both CSTs were observed around both lateral ventricles due to bilateral ventriculomegaly versus control subjects (70
89 years old female, white arrow) and a discontinuity of both tracts to the bilateral cortex was apperent
(white dash circle) (b) control. (C) (a) The right CST did not extend to the right cortex (yellow arrow)
and (b) the left CST spread out (green dash arrow). (D) CSTs of a control subject. (E) (a) The right CST
(red) was posteriorly shifted at the corona radiata (orange arrow) and the left CST (yellow) was spread out as compared with control subjects (white dashed circle). (b) Left and right CSTs of a control
Note: R, right; L, left; A, anterior; P, posterior.
95 Table 1. DTT parameter values of the corticospinal tracts of the patient and control subjects. Note: SD:, standard deviation,; CST, corticospinal tract; FA, fractional anisotropy; TV, tract volume; MD, mean diffusivity. **Parameters were two SDs above or below mean normal control subject values.

Ans> as your comment, we revised manuscript,

Discussion
The corticospinal tract constitutes the main white matter motor pathway  [8]. In this patient with generalized motor weakness after SAH, the CST was investigated using DTI/DTT MR imaging. Bilateral CSTs reconstruction using DTT revealed side-dependent differences between are subject in comparison to a matched cohort of control patients.  More specifically, changes in CST structures were observed around the corona radiata and lateral ventricle; the right CST had moved posteriorly and
the left CST was spread out at the level of the corona radiata, compared to healthy controls. Apart from displacement, the CSTs revealed signs of discontinuation at the cerebral cortex. Furthermore, in the left CST, FA and
MD values were significantly elevated and TV values significantly reduced, and in the right CST, MD values were significant alerted (in what manner?).

Ans> as your comment, we revised manuscript,

Ans>(in what manner? -->Furthermore, in the left CST, FA and MD values were elevated and TV values reduced, and and in the right CST, MD values were alerted due to the more or lower difference than 2 standard deviations compared with control subjects.

FA values represent degrees of directionality at a microscopic level [9–11], and thus, provide a means of assessingmicrostructural integrity of axons, myelin, and microtubules [4,12]. On the other hand, MD values provide quantitative measures of water diffusion, and are are indicative for pathological changes taking place in white matter [13]. Increases in MD represents vasogenic edema or axonal damage whereas a decrease of MD values indicate neural injury [19]. TV values represent numbers of voxels within neural tracts [14]. Therefore, FA and TV reductions in combination with a MD increases, indicate the presence of neural injury [6,14].
In a previous study, it was reported that patients with normal pressure hydrocephalus (NPH)
had initial high FA values, explained by mechanical compression and decreased FA and increased MD values as a result of secondary degenerative changes.  [3,15,16].

Ans> as your comment, we revised manuscript,

In this case, our patient exhibited the bilateral displacement and discontinuation of the CSTs before reaching the cerebral cortex. In addition, our MR imaging revealed high FA and MD values in combination with low TV value in the left CST. The FA value of the right CST however, was not increased significantly. Based on these findings, we hypothesize the mechanism of motor weakness in our case to be different between sides. At week 4 after onset, posterior displacement of the right CST, as the result of hydrocephalus caused ventriculomegaly, resulted in a discontinuation of the tract to the cerebral cortex. This while significantly high MD values were found with the number of voxels contained within neural tracts
and FA values remaining unchanged. We believe that the changes to the right CST are mainly caused by vasogenic edema without neural injury, leading to the discontinuation of the tract resulting in loss of left motor function. In contrast to our results during the subacute phase, aprevious study reported the formation of vasogenic edema following SAH in the acute phase [20]. Further MRI assessment of the timely development of vasogenic edema after SAH is necessary. 

Ans> as your comment, we revised manuscript,

However, in the left CST, a spreading within the corona radiata was observed due to ventriculomegaly caused mechanical compression. This, based on the observed increased FA values, decreased TV
and increased MD indicative of degenerative change or neural tract injury. These observations lead to the paralysis of right limbs.

However, we were able to exclude age-
FA TV MD ( ⅹ 10-3mm2/s)

(please clarify this age correction further)
Patient
Right 0.632 3106.000 0.879**
Left 0.666** 2148.000** 0.763**
Controls mean (SD) (n=7) 0.616 (0.017) 3002.286 (451.384) 0.709 (0.019)
Formatted: Indent: Left: 0 cm, Right: 0 cm
associated microstructural changes because of no reduced FA value in a patient [17].

Ans> as your comment, we revised manuscript,

And we edited Table 1.

This next section tries to explain the side difference in motor function, however it is unclearly written and the exact message does not come across. 

Ans> Additionally, as a previous study, the analysis of CST for upper and lower limbs using DTT is difficult due to the discontinuation of CSTs at the cerebral cortex [18].

Additionally, regardless of the different motor power between upper and lower limbs, the exact anatomic location analysis of CST using DTT is difficult due to the discontinuation of CSTs at the cerebral cortex [18]. However, we believe that a few fibers to the cerebral cortex could play a role in upper motor function and tracts associated with lower motor function were injured due to a compression of lateral ventricle and their discontinuation. For However, the defined causes of mechanical compression on the left side there and vasogenic edema in the right side were not defined effect at 4 weeks after SAH,. the further study was required. Therefore, at 4 four weeks after onset (subacute phase), injury of the left tracts resulted from 140 degenerative change of tracts (increased MD value and decreased TV value) and still mechanical compression (increased FA value). Damage to the right CST caused vasogenic edema by ventriculomegaly. We believe that because of the larger left ventricle, mechanical compression had greater influence on the left CST in terms of increasing FA. In other words, we indicate that mechanical compression and degenerative changes by ventriculomegaly can coexist and induce tract injury. In addition, we believe that the right CST injury results from vasogenic edema based on differences in morphology of the tract caused by difference in ventricular size.

Ans> as your comment, we revised manuscript,

The most feared complication following SAH is delayed cerebral ischemia. To prevent delayed cerebral ischemia, our patient was treated with aspirin and cilostazol [22]. Another complication is hydrocephalus [23]. The patient received an EVD during the acute phase and was constantly being monitored for persisting hydrocephalus. Previous studies have also presented mechanisms of motor weakness after SAH or NPH [3,15]. It has been suggested that one of mechanisms of paraplegia or paraparesis after SAH was hydrocephalus [3]. However, to the best of our knowledge, this case report is the first to observe side-dependent different mechanisms resulting in motor weakness as a result of asymmetrical hydrocephalic ventriculomegaly (mechanical compression at the  left hemisphere and vasogenic edema at the right hemisphere).

Ans> as your comment, we revised manuscript,

However, this study has some limitations including: (1)The external validity of these observation remains limited because they are based on a single case, demanding larger scale long-term studies are to confirm our hypothesis (2) because our study was in the subacute phase, long-term follow up is necessary. Additionally, the long-term image follows up for the determination of motor recovery and the change of ventricle size for hydrocephalus is being required (what does this sentence try to say?); (3) as the above mention, because causes of mechanical compression in the left side and vasogenic edema were not defined, the further study using large scale is required; (this is repetition of the same argument) (4) DTT interpretation is operator-dependent leading to potential performance bias [8]; and (5) Finally, further clarification via eletrophysiological examination of long fiber tracts would have been helpful but was hindered due to the existing craniotomy defect [21].  

Ans> as your comment, we revised manuscript,

Ans> this is repetition of the same argument:==> à erase sentence

Ans> long-term image follows up for the determination of motor recovery and the change of ventricle size for hydrocephalus is being required (what does this sentence try to say?);--> à this mean is long-term follow up. Thus, we erase this sentence due to the same sentence “ long-term follow up is necessary.”

Conclusion
Although DTI is subjected to operator-dependency [8], we observed the coexistence of side-dependent compressive and degenerative damage to the CST cause by either ventriculomegaly or direct compression leading to bilateral motor weakness. Here, DTT proves to be a useful tool in assessing the mechanism behind post-SAH persisting motor weakness

 Ans> as your comment, we revised manuscript,
